# A Web-Based Application to Monitor and Inform about the COVID-19 Outbreak in Italy: The {COVID-19ita} Initiative

**DOI:** 10.3390/healthcare10030473

**Published:** 2022-03-03

**Authors:** Corrado Lanera, Danila Azzolina, Francesco Pirotti, Ilaria Prosepe, Giulia Lorenzoni, Paola Berchialla, Dario Gregori

**Affiliations:** 1Unit of Biostatistics, Epidemiology and Public Health, Department of Cardiac, Thoracic, Vascular Sciences, and Public Health, University of Padova, 35131 Padova, Italy; corrado.lanera@unipd.it (C.L.); danila.azzolina@unife.it (D.A.); ilaria.prosepe@studenti.unipd.it (I.P.); giulia.lorenzoni@unipd.it (G.L.); 2Department of Environmental and Preventive Sciences, University of Ferrara, 44121 Ferrara, Italy; 3Department of Land, Environment, Agriculture and Forestry, University of Padova, 35020 Padova, Italy; francesco.pirotti@unipd.it; 4Department of Clinical and Biological Sciences, University of Torino, 10043 Torino, Italy; paola.berchialla@unito.it

**Keywords:** COVID-19, web application, shiny app, monitoring tool

## Abstract

The pandemic outbreak of COVID-19 has posed several questions about public health emergency risk communication. Due to the effort required for the population to adopt appropriate behaviors in response to the emergency, it is essential to inform the public of the epidemic situation with transparent data sources. The COVID-19ita project aimed to develop a public open-source tool to provide timely, updated information on the pandemic’s evolution in Italy. It is a web-based application, the front end for the eponymously named R package freely available on GitHub, deployed both in English and Italian. The web application pulls the data from the official repository of the Italian COVID-19 outbreak at the national, regional, and provincial levels. The app allows the user to select information to visualize data in an interactive environment and compare epidemic situations over time and across different Italian regions. At the same time, it provides insights about the outbreak that are explained and commented upon to yield reasoned, focused, timely, and updated information about the outbreak evolution.

## 1. Introduction

The COVID-19 pandemic outbreak posed several questions about transparency and access to data and the need to provide information easily understandable to a broader audience than the scientific community [1].

Strict surveillance of epidemics is essential to inform health policy-makers about the actions to be undertaken [2]. On the one hand, the need for the adoption of community-based containment strategies (e.g., social distancing measures, lockdown), together with strict surveillance of infectious disease units and ICUs (Intensive Care Units) in healthcare facilities, is noteworthy [3]. On the other hand, tracking the epidemic is essential to assess the effectiveness of the adopted policies [4]. Given the effort required for the whole population to adopt containment measures and appropriate behaviors, the importance of informing the public of the epidemic’s evolution emerged immediately. All stakeholders, i.e., health policy-makers and planners, the scientific community, and the general public, should have access to transparent data and should be put in a position to correctly understand and use them both for their analysis and to reproduce or improve the data reported. Transparency in communication, beyond a rhetorical commitment, is also essential to ensure that the perception of risk aligns with the actual risk [5].

The web-based dashboard developed at the Center for Systems Science and Engineering at John Hopkins University was one of the first of several online interactive platforms devoted to tracking the novel coronavirus [6]. Several web-based platforms monitor the pandemic at a national or territorial district level, and some of them also show raw data and forecast the epidemic evolution [7,8,9]. However, these dashboards are not tailored to communicate the reported results to the general public in plain language; graphs, maps, and tables should be easily understood and correctly interpreted by the public to be effective, reduce uncertainty and facilitate increased risk awareness [10].

The lack of such aspects of communication pertaining to risk-related information tends to undermine public compliance and acceptance of the ethical rationale for adopting containment measures, potentially limiting individual freedom [11].

This work proposes an open-source, web-based tool facilitating the communication of the fundamental epidemiological COVID-19 indicators to the general public and experts in this research field. The achievement of this purpose is carried out by offering to web-app users dedicated sections and insights enriched with explanations and supports for the understanding of the reported indicators.

## 2. Materials and Methods

The COVID-19ita project is a web-based application developed and first released in March 2020, serving as the front end of the eponymously named R package freely available on GitHub (https://github.com/UBESP-DCTV/covid19ita (accessed on 20 February 2022)).

The web-based application is deployed both in Italian and English. The two instances are hosted and freely served by the computational facilities at the Unit of Biostatistics, Epidemiology, and Public Health of the University of Padova and are available online at https://r-ubesp.dctv.unipd.it/shiny/covid19ita/ (accessed on 20 February 2022) and https://r-ubesp.dctv.unipd.it/shiny/covid19italy/ (accessed on 20 February 2022), respectively.

The working structure of the web app together with the existing relationships with the various data sources and development environments are described in Section 2.1.

The overall app structure (Appendix A) manages the COVID-19 official data (Section 2.2, Data), the pandemic indicators (Section 2.3, Indicators), and the COVID-19ita R package providing the implementation of the web application with a directly usable R version of the official data (Section 2.4, R-Package). The data sources are described together with the indicators and the R package.

### 2.1. WEB App Structure

The R Shiny web app is freely hosted on https://r-ubesp.dctv.unipd.it/shiny/covid19ita/ (accessed on 15 February 2022). The web application contains an open-source code stored on a GitHub UBEP repository (Unit of Biostatistics, Epidemiology and Public Health, University of Padua, Italy). Everyone can access the code placed on this repository; for example, General Public, the clinical staff, and the government.

A local UBEP server automatically performs scheduled, daily operations after government data is published. These operations are useful to automatically update the data and analyses reported in the shiny web application. These automated procedures consist of:(1)Upload any changes that have been saved to the UBEP’s Github repository(2)Pull the daily Updates concerning the COVID-19 data(3)Save the log of the operations performed on the web app code and system on my SQL database.

UBEP researchers maintain the system and push changes to the local Github repository. They also take care of producing results on the web app tailored to both the general public, clinical experts in the field, and governmental stakeholders. The structure is represented in the flowchart in the Appendix A.

### 2.2. Data

The project directly pulls data from the official records that track the Italian COVID-19 outbreak on a national, regional, and provincial level. In Italy, since 24 February 2020, official data were processed initially and made available by the Presidenza del Consiglio dei Ministri—Dipartimento di Protezione Civile (Italian Civil Protection Department, http://www.protezionecivile.it/web/guest/home/ (accessed on 20 February 2022)) and licensed under CC-BY-4.0 (https://creativecommons.org/licenses/by/4.0/deed.en/ (accessed on 20 February 2022)) as provided by the Ministero della Salute (Ministry of Health, http://www.salute.gov.it/ (accessed on 20 February 2022)).

The Italian COVID-19 Surveillance Data are published on a GitHub national repository (https://github.com/pcm-dpc/COVID-19 (accessed on 20 February 2022)). The official daily data flow on COVID-19 starts at the regional level to arrive at the Civil Protection Department following this process on a daily basis [12]:The Italian regions compile the data by 16:30 on a web application provided by the National Health Institute (Istituto Superiore di Sanità).The Ministry of Health receives the data, performs a data quality control procedure, and then sends back the records to the Department of Civil Protection by 17:00.The Department of Civil Protection performs another data quality control procedure by 18:00, processes the data, and eventually publishes the database on GitHub.The official data web portal also provides ex-post control through a specific GitHub issue module that can be opened by the community. As a consequence of this review process, the official data on epidemic trends in Italy can be subject to retrospective reviews and updates.

Data concerning overall mortality during the COVID period have also been considered and retrieved by the Italian National Institute of Statistics (ISTAT) data source (https://www.istat.it/it/archivio (accessed on 20 February 2022)).

### 2.3. Indicators

The main COVID-19 indicators provided by official Italian data sources daily used for the platform are (1) the number of COVID-19 hospitalizations with symptoms; (2) the number of COVID-19 hospitalizations in intensive care; (3) the number of COVID-19 hospitalized patients; (4) the number of persons in home isolation; (5) the total currently positive (hospitalized + home isolation) subjects; (6) the daily changes in total positives (total positive current day—total positive previous day); (7) the number of new current positives (total current day cases—total previous day cases); (8) the number of discharged/cured persons; (9) the number of deaths; (10) the number of total positive cases; (11) the number of swabs processed by molecular testing; (12) the number of subjects tested, and (13) the daily ICU admissions.

### 2.4. R-Package

Keeping track of the official data flow and reviews could be difficult. For this reason, the whole COVID-19ita project was developed under the GitHub version control system. This GitHub facility allows the project to have access to different versions of the package and data, where each version represents a snapshot of the whole project at past time points [13].

The COVID-19ita R package (https://github.com/UBESP-DCTV/covid19ita (accessed on 20 February 2022)) provides the implementation of a Shiny application (https://shiny.rstudio.com/ (accessed on 20 February 2022)) together with a directly usable R version of the official data. The package was developed with the golem (https://github.com/ThinkR-open/golem (accessed on 20 February2022)) package framework for building production-grade shiny applications. The covid19ita package defines and collects all the functions used and useful to run the corresponding Web application. This R package is also available as a GitHub public repository [13]. The data used for analysis and computations within the platform are continuously updated and supplied within the COVID-19ita R package.

The application R code is structured into separate modules, one for each component of the web platform. Modular programming enhances the programming code readability, facilitating collaboration between multiple scientists and allowing easy adaptation of the code without the need for extensive reprogramming [14].

## 3. Results

The results are structured in two parts; the first one, referred to the web application, describes the leading functionalities of the different app sections. Some examples have been reported from the (1) Highlights; (2) Epidemic; (3) Epidemiological indices, (4) Maps, and (5) Forecasting tools sections of the web app.

The second part reports the number of accesses to the app with the relative territorial origin.

### 3.1. Web Application

The COVID-19ita (https://r-ubesp.dctv.unipd.it/shiny/covid19ita/ (accessed on 20 February 2022)) web interface provides a single-page multi-dashboard interface to data, trends, annotated models, and insights for the COVID-19 outbreak in Italy.

The web page interface is composed of three main parts: (i) a header section hosting the title and a dropdown menu to access quick side news and to switch languages; (ii) a sidebar hosting the links to navigate across the application parts and a set of primary pandemic outbreak metrics (i.e., the overall number of tests performed, confirmed, active hospitalized, and recovered cases, and deaths); (iii) the main body, in which the application displays the selected content.

The platform reports the indicators aggregated at the national level or by region. For the provinces, the web application reports only the overall number of cases, which is the only epidemic indicator provided from official data sources for this kind of geographical classification.

The web app reports a home page section containing bibliographic references and a brief description of the working group; there is also a link dedicated to the web app user for feedback and bug communication.

Highlights. The highlights section investigates the epidemic outbreak in depth by providing the results of ongoing analyses, such as the estimation of the likely effect of the health policies implemented to contain the spread of COVID-19.

In this section, all analyses reported are explained and discussed, paying attention to the risk communication of public health emergencies, reflecting a framework of user-centered design [15].

Risk communication is performed on the platform by specifying, in dedicated sections, all the assumptions of the estimated models and forecasting scenarios in a language that can be understood even by users who are not experts in epidemiology and biostatistics.

Some examples reported in the highlights section are as follows:(1)Analysis of mortality in Veneto. The excess mortality estimation was performed on the Veneto region during the early phase of the pandemic by comparing the mortality by territorial levels (provinces) in March 2020 with the same time frame of the same years (Figure 1). The analysis of COVID-19 mortality in Veneto has also been published elsewhere [16].(2)Effect of swab policy. To assess the potential impact of the swab policy, the web tool compares the Veneto and Piedmont regions, which followed different policies. In Italy, during the early stages of the pandemic, Veneto adopted a more broad-based testing policy than Piedmont. The Veneto swab model has been applied to Piedmont to predict hospitalizations. In particular, Figure 2 shows that based on the total number of cases, Piedmont and Veneto should have approximately the same number of hospitalizations: on the top side of the figure, this is represented by the two red and green curves, which are almost superimposed. However, if we also visualize the real data of the hospitalizations (red circles) on the graph, we observe that in Piedmont, the hospitalizations are higher than those in Veneto. The plot reported on the bottom side of the web page demonstrates that, including as a covariate in the model the number of swabs performed over time, it is possible to observe that the Veneto model predicts with a good approximation the hospitalizations in Piedmont, thus explaining the observed difference among regions (Figure 2). A comparative analysis between the Veneto and Piedmont regions has also been reported in the literature [17].(3)Preliminary impact of the containment policy in Veneto. The web app reports a first-look impression of the possible effect of the health policies implemented in Veneto to contain the spread of COVID-19 during the early phases of the pandemic. The plot reported in Figure 3 compared the predictable trend based on data as of March 3 with the trend observed in the Veneto region to understand whether some or all of the actions implemented had a plausible effect of slowing down the evolution of the epidemic. The figure shows a slowdown after 2 March, the day on which a changepoint in the epidemic trend was observed. Based on this comparison (estimated curve on 2 March and data observed in the following days), it was possible to estimate some indicators, for example:The number of positive cases avoided as of 12 March in Veneto is 348 (95% CI = 322–373);A slowdown of the epidemic to 12 March is equal to 15.91 cases/day (95% CI = 11.99–19.82), with a peak on 6 March of 40 cases/day fewer than expected. The same analysis was performed on the Veneto hospitalization data as reported elsewhere [18].

Epidemic. The “Epidemic” section provides data on the event time trends on a national, regional, and local level with measures provided in both cumulative and incidence distributions, with an option to select a linear (the default) or logarithmic scale.

For each region, the app provides an interface to compare the different indicators on the same plot. The web application is tailored to compare the same indicator across multiple regions in the same plot environment (Figure 4). Figure 4, for example, represents the ICU admissions in Veneto and Piemonte regions.

Epidemiological Indices. The “Epidemiological Indices” section provides an overview of the leading epidemiological indices (e.g., number of cases over the number of tests performed). The computed values are reported within a live and dynamical plot environment; the web app user also offers the possibility of selecting the regions for which to report the pattern of the selected indices within comparative graphs (Figure 5).

In the figure (Figure 5), the *x*-axis shows the percentage of patients not hospitalized who underwent COVID-19 testing; the *y*-axis indicates the percentage of ICU admissions in the resident population. The continuous line indicates the local polynomial regression smoothing. In the early phases of the pandemic, the plot reports a higher percentage of asymptomatic/mild symptomatic patients tested since the beginning of the outbreak in Veneto, corresponding to a lower percentage of subjects admitted to the ICU in comparison with Lombardia [19].

Maps. The “Maps” section shows a graphical representation of the epidemic patterns by geographical area. Data are represented on the map with provincial details. The indicators considered for the map representation are as follows:Total COVID-19 cases in absolute value and over a population of 10,000 residents;Daily cases in absolute value, over a population of 10,000 residents, and on a standardized scale; andThe average number of cases over a seven-day window on absolute and standardized scales.

Once the indicator has been selected, it is possible to drag the slider above the map panel (Figure 6) to dynamically change the day and color over the date range (currently from Monday, 24 February 2020, to the last available date).

The application simulates a time-lapse of the spatial distribution of COVID-19-positive cases over the Italian provinces; the darker-colored areas indicate a higher incidence of COVID-19 cases on the selected day (Figure 6, Panel A). It is also possible to switch the color palette and convert the indicators from linear to logarithmic scales to improve visualization when the frequency distribution of values is asymmetric/skewed.

When the flag “Fixed” is selected, the map scale will not change when changing the date, but the system considers the full range calculated from all values of the chosen variable across the full period. If the flag is unchecked, the scale will change every time a different day is chosen according to the selected day’s values (Figure 6).

By selecting the field “Labels”, it is also possible to show the values reported on the maps during computation (Figure 6, Panel B).

Intensive Care Unit. A smoothing state space model (ETS) estimation approach is used to predict the series of COVID-19 ICU admissions on the web platform. The estimates with 10-day forecasts are considered by performing the calculation at the regional level.

The predictive tool has also been reported by the National Agency for Regional Health Services (AGENAS) website (https://www.agenas.gov.it/covid19/web/ (accessed on 20 February 2022)).

The forecast estimates were calculated using an Exponential Time Series (ETS) smoothing model [20].

The ETS model considers the Error, Trend and Seasonal components of the time series; the model components can be additive (A), multiplicative (M), or none (N). The damped trend (Ad) has been also considered as a possible parametrization that “dampens” the trend to a flat line in the future horizon time [21].

The suitable model parametrization, consisting of the combination of ETS, is automatically computed according to the minimum of BIC criterion [22]. Our estimates, together with the optimal parametrization, are automatically updated when new data are obtained on a daily basis from the Italian Civil Protection Department.

The forecasting instrument has been developed in a dynamic environment: the web app user can select the time frame for which the estimation model is performed.

Figure 6 presents an estimation example: once the slider has been moved to the left, the model is estimated until 11 November 2020, and the forecast estimates are shown for the next 10 days (Figure 7, Panel A).

By moving the slider to the right, up to the last update date, it is possible to obtain the model estimated at the last available data with the forecasting going on the next 10 days (Figure 7, Panel B). The ICUforecast error (Figure 7, Panel C) is plotted and calculated (as squared error score [23]) on increasing fractions of the historical ICU time series. The Figure shows that the error stabilizes downwards after six months of pandemic data.

### 3.2. Web Application Number of Connections

From 14 March 2020 (initial on-line date) to 31 December 2021, 25,641 unique visitors explored the platform 53,388 times, for a mean time of 2′17″/session. Users visited the platform from many countries, e.g., Italy (93.39%), USA (1.47%), France (0.75%), Germany (0.44%), UK (0.43%), Spain (0.35%), Switzerland (0.29%), the Netherlands (0.24%), Denmark (0.16%) and others (0.22%). The vast majority of sessions were on mobile devices, i.e., Android (60.64%), iOS (39.13%), and other (0.04%), while connection from desktops/laptops were rare, i.e., Windows (0.19%).

## 4. Discussion

Several open-data web sources have been developed for the COVID-19 epidemic worldwide and in Italy. Most of them have been developed to encourage network connections among epidemiologists, data scientists, statisticians, and researchers involved in COVID-19 epidemic research.

In Italy, for example, it is possible to identify the principal open-data sources and web platforms pertinent to the study of COVID-19. The Civil Protection Department updates a GitHub repository daily (https://github.com/pcm-dpc/COVID-19 (accessed on 15 February 2022)) with data organized by regions and provinces. Moreover, the GEDI *Gruppo Editoriale* provides a web portal where the COVID-19 data are organized in plots (https://lab.gedidigital.it/gedi-visual/2020/coronavirus-in-italia/ (accessed on 18 February 2022)). The National Statistics Institute (ISTAT) also reports the analyses and the data on mortality during the COVID pandemic period (http://dati.istat.it/Index.aspx?QueryId=19670 (accessed on 20 February 2022)).

In general, these web-based COVID-19 monitoring platforms describe the pandemic evolution data while also providing forecasting of the epidemic’s evolution [7,8,9]. However, the dashboards and results included in these applications are not tailored to communicate to the general public. In this regard, our web application proposes the dissemination of updated knowledge not only presenting the pandemic time series and indicators but also including some sections aimed at explaining statistical-epidemiological phenomena that are difficult to convey properly by the mass media and communication agencies.

For example, the web app reports some specific in-depth sections that explain some epidemiological concepts, such as the possible impact of the more- or less-widespread testing policies on COVID-19 hospitalizations [18,19]. Furthermore, for example, in Italy, during the early stages of the epidemic, the mass media often spoke of the identification of the pandemic peak as a rough point from which the effects of the pandemic would begin to subside. The web app in a special tab is aimed at communicating the complexity of such a forecast by showing the public the type of parameters and assumptions involved in the estimates. Other specific tabs are provided to communicate the first possible impacts of the containment policies implemented in the Veneto region and in other regions to contain the COVID-19 epidemic outbreak during the first wave of the pandemic. These monitoring tools could be useful for the decision-maker to assess the short-term effects of the prevention policies, but they are also useful to motivate the citizen who quantifies the effects of the adherence to the policies in terms of the number of hospitalizations or deaths avoided [18].

Moreover, the Map instrument tool reported in this web application gives a representation of COVID-19’s spread across geographical locations (provinces); the application provides a day-by-day animation of the progression of COVID-19 diffusion. Additionally, the GEDI web application provides some maps concerning the state of contagion in the Italian territory. However, the aforementioned instrument is not dynamic; only the last-available data can be shown at the regional territorial level.

In general, our web application provides graphical insight into the data accompanied by explanations and interpretative support about the outbreak to facilitate the communication of data and indicators to all target users: health policy-makers and planners, the scientific community, and the general public. In Italy, other web apps propose the results of analyses on the official data, providing explanations of the results for communication purposes—for example, the Lab24 platform of the Sole 24 Ore communication agency (https://lab24.ilsole24ore.com/coronavirus/ (accessed on 20 February 2022)).

All of these applications, including the last one, have not been developed in a GitHub repository; instead, our COVID-19 monitoring tool is an R-Shiny web application developed in the GitHub environment. This development system for the COVID-19 application is also used on comparative COVID-19 data and analyses at the international level (https://mahdisalehi.shinyapps.io/Covid19Dashboard/) (accessed on 21 February 2022), although this application does not report explanations and epidemiological insights aimed at understanding the impact of the epidemic.

This R-Shiny programming with the development approach in GitHub has been used by the authors of the aforementioned web app to ensure a user-friendly open-source R Shiny software guaranteeing a high sense of transparency and reproducibility of codes, data, and results with the general public and scientific community [24].

In addition to the code-sharing-oriented development environment, our web app aims to communicate epidemiological concepts and COVID-19 monitoring results to nonexpert audiences. In general, the lack of knowledge of an epidemiological phenomenon in an emergency could inspire nonrational behaviors that are difficult to manage in a situation or crisis [25]. During the early stages of the pandemic, for example, there were episodes of collective fear in Italy, such as runs on supermarkets and uncontrolled movements in different areas of the country. Nonetheless, the literature has also shown strong psychological effects of the pandemic in situations of uncertainty, such as increased anxiety, depression, and psychological distress [26]. In this context, spreading up-to-date knowledge, including explanations about the state of the epidemic, could be crucial [27].

### 4.1. Limitations

The web app was designed to facilitate the communication of risk and COVID-related epidemiological concepts to the general public. However, we do not have results and indicators that indicate how much the web app has impacted public awareness of the phenomenon. Further research developments are needed in this regard.

In addition, a potential side effect of public access to the analysis and monitoring instruments of epidemic trends could be a communication gap that may instill anxiety in the general public if the results are misinterpreted [28]. This is true especially if the information is improperly conveyed by the mass media and communication agencies [29].

From an IT point of view, currently, a single instance of the application is running for all web app users. This aspect could slow down the user experience regardless of whether many users connect to the app simultaneously. However, the web app programming structure will be easily scalable to a containerized multi-process version running distinct instances of the application for each user.

### 4.2. Future Research Developments

Given the impact of the web-based application on both the general public and of those in charge of decision-making, we aim to maintain and further develop the web-based tool to closely monitor the epidemic outbreak in our country to provide a means for the correct understanding and interpretation of the phenomenon, as well as support to facilitate and foster cooperation among scientists.

## 5. Conclusions

The web application pulls the data from the official repository of the Italian COVID-19 outbreak at the national, regional, and provincial levels. The app allows the user to select information to visualize data in an interactive environment and compare epidemic situations over time and across different Italian regions. The website includes plots and explanatory sections dedicated to communicating pandemic indicators and analyses to the general public, experts in the epidemiological fields, and stakeholders.

## Figures and Tables

**Figure 1 healthcare-10-00473-f001:**
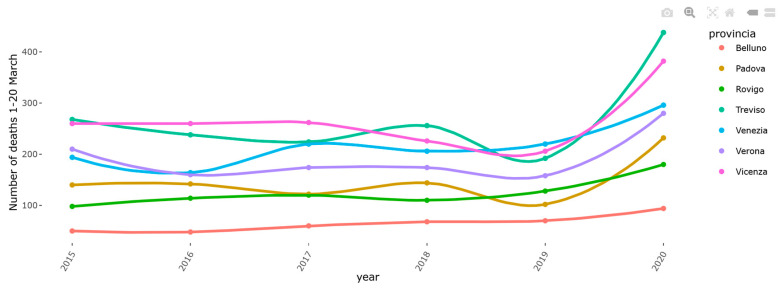
One example (web page screenshot) of the content in the Highlights section (namely, changes in mortality by province in the Veneto region).

**Figure 2 healthcare-10-00473-f002:**
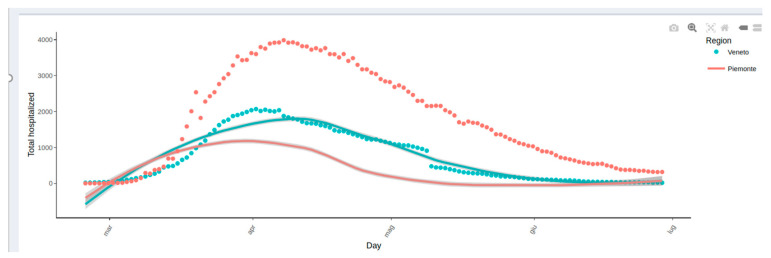
One example (web page screenshot) of the content of the Highlights section (namely, the impact of testing on hospitalization) as displayed in the main body of the COVID-19ita application. The figure represents the Veneto model applied to the Piedmont data without taking into account the information regarding the number of swabs performed. In Piedmont (red curve) the number of expected hospitalizations according to the model should be similar to that of Veneto (green curve), but it differs considerably from the observed data (red dots).

**Figure 3 healthcare-10-00473-f003:**
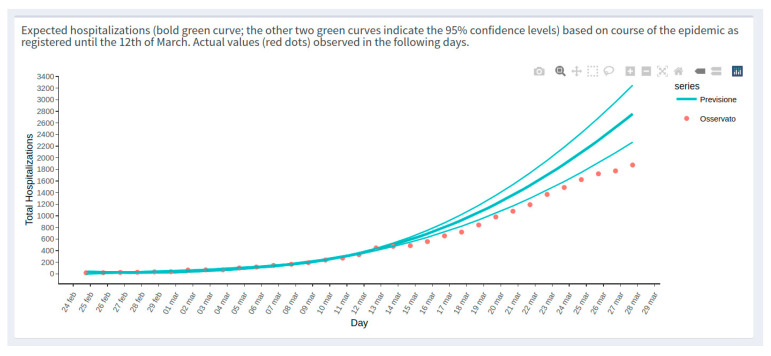
One example (web page screenshot) of the content of the Highlights section (namely, the possible effect of the health policies implemented by the Veneto region) as displayed in the main body of the COVID-19 pandemic application.

**Figure 4 healthcare-10-00473-f004:**
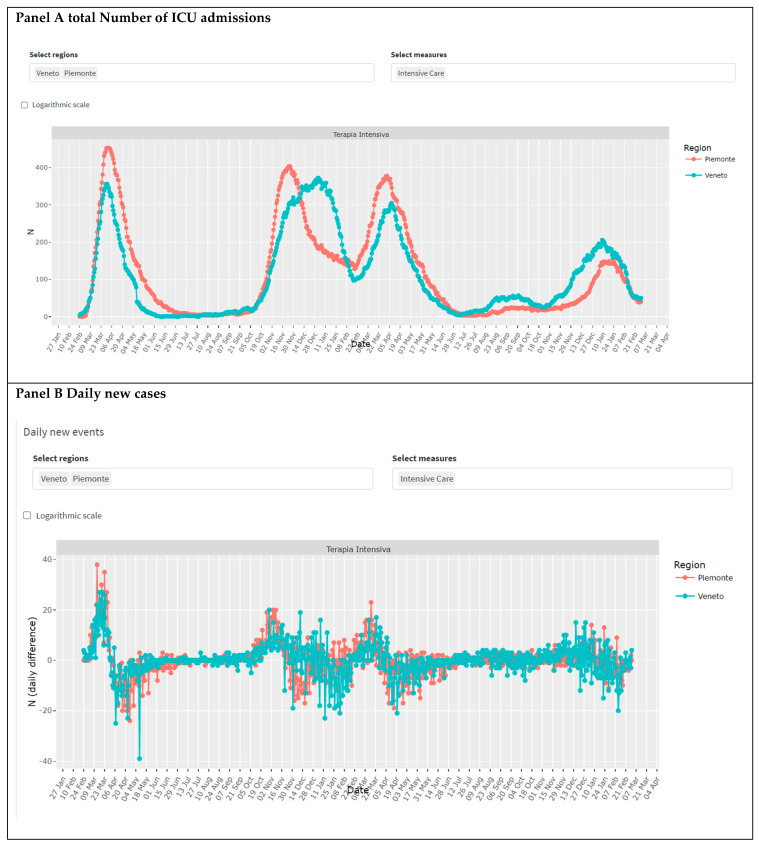
Epidemic section of the COVID-19 pandemic application (web page screenshot). The first two sections from the top report the total number of ICU cases (Panel (**A**)) and the daily ICU admissions (Panel (**B**)) in the Veneto and Piemonte region. Region and indicator can be chosen by the user.

**Figure 5 healthcare-10-00473-f005:**
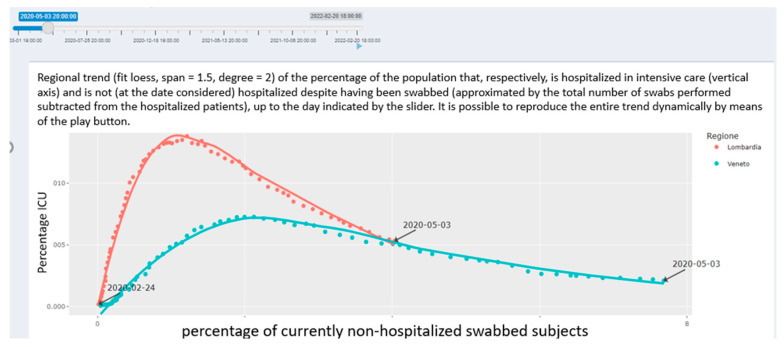
Section for leading indices provided by the COVID-19 pandemic application (web page screenshot). In this reported example, the dynamics across time of the percentage of people tested against the percentage of people admitted to the ICU are displayed for the selected region (in the example shown: Lombardia and Veneto) both as a time-dynamic running graph and as tabular data.

**Figure 6 healthcare-10-00473-f006:**
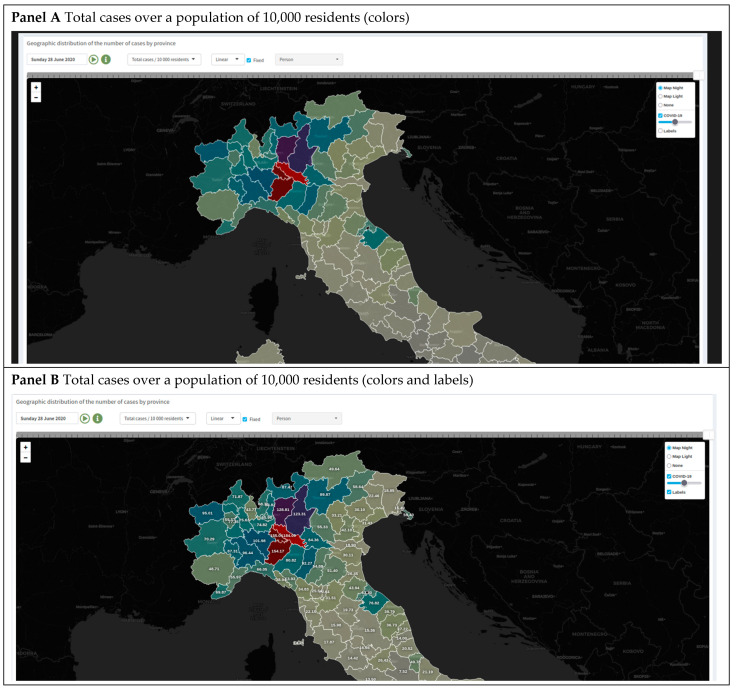
Map section of the COVID-19 website app (web page screenshot). The map shows regional or provincial tessellation for the nation. Each tile is colored proportionally to the intensity of the selected measure. The option to run a day-by-day animation of the progression of the chosen measure is also provided. Panel (**A**): Map representation with a color palette. Panel (**B**): Map representation with labels.

**Figure 7 healthcare-10-00473-f007:**
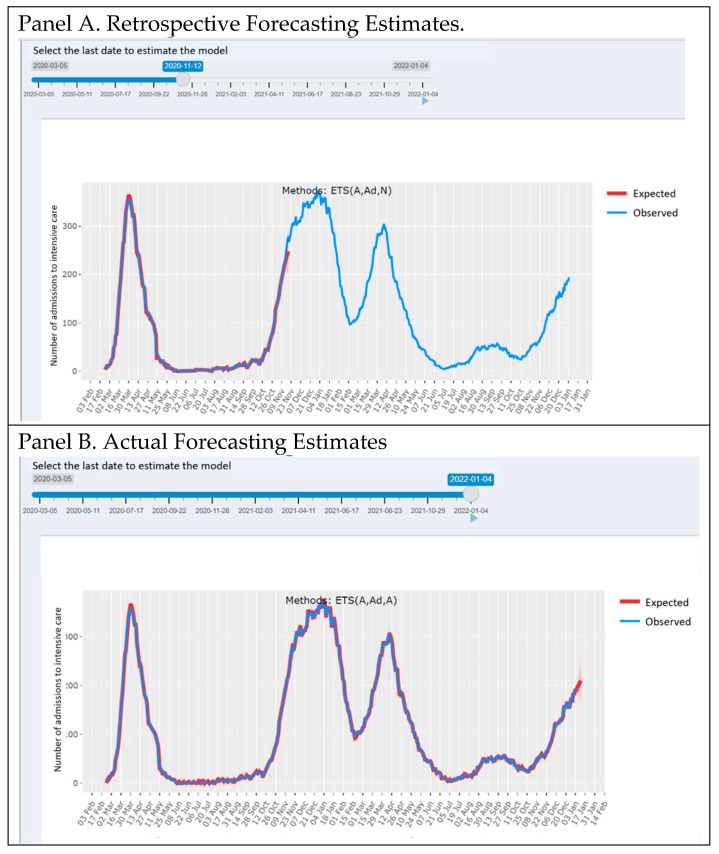
ICU forecasting tool of the COVID-19 advent app (web page screenshot). Panel (**A**): Retrospective forecasting estimates obtained by moving the slider toward the 11th of November. In red is the estimated ICU occupancy, and in blue are the observed ICU admissions. The red-shaded area indicates the 95% forecasting confidence intervals. Panel (**B**): Actual forecasting estimates obtained by moving the slider toward 11 November. In red is the estimated ICU occupancy, and in blue are the observed ICU admissions. The red-shaded area indicates the 95% forecasting confidence intervals. Panel (**C**): Forecasting error (squared error score) according to the forecasting time window (black dots); the blue line represents the local regression smoothing estimate (LOESS).

## Data Availability

Linkages to official Covid-19 Italian data data used for computations are available at repository https://github.com/UBESP-DCTV/covid19ita (accessed on 20 February 2022).

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
