# Peer review of "A Web-Based Application to Monitor and Inform about the COVID-19 Outbreak in Italy: The {COVID-19ita} Initiative"

_healthcare, 2022, doi:10.3390/healthcare10030473_

Round 1
Reviewer 1 Report
This article develops a public open-source tool to provide information and explanations on the evolution of the COVID-19 outbreak in Italy. The motivation is good and the topic is very interesting. However, I don’t agree to accept the manuscript in the current version. Frankly speaking, I can’t find any novel idea on computing model and algorithm. It seems that the authors only present a tool using the given data, and describe some details on how it works.
Here comes my suggestions.
- How to get the forecasting estimates? Please explain the specific computational tricks.
- How well does this tool work in a worst-case scenario. For instance, you can’t get the timely data, and in this case, will the prediction succeed or fail?
- What makes your tool work effectively? Which types of parameters determine the efficiency of prediction, and which of these parameters are decisive?
- Please improve your written. For example, the reference should write in a suitable place, not at the beginning of a sentence.
Author Response
We would like to thank the reviewers for their comments that helped to improve the manuscript quality. The rebuttal letter has been prepared with point-by-point responses. The active track has been turned on in the manuscript to highlight the changes
Reviewer 1
This article develops a public open-source tool to provide information and explanations on the evolution of the COVID-19 outbreak in Italy. The motivation is good and the topic is very interesting. However, I don’t agree to accept the manuscript in the current version. Frankly speaking, I can’t find any novel idea on computing model and algorithm. It seems that the authors only present a tool using the given data, and describe some details on how it works.
- How to get the forecasting estimates? Please explain the specific computational tricks.
We agree with the reviewer; other details have been included in the Intensive Care Unit forecast description section.
The forecast estimates were calculated using an Exponential Time Series (ETS) smoothing model. The estimates have been also published on the official platform of the Italian agency AGENAS (National Agency for Regional Health Services) (https://www.agenas.gov.it/covid19/web/index.php?r=site%2Findex).
- How well does this tool work in a worst-case scenario. For instance, you can’t get the timely data, and in this case, will the prediction succeed or fail?
We agree with the reviewer's concerns. Our estimates are automatically updated when new data are obtained on a daily basis from the Italian Civil Protection. Specifications are included in the manuscript in the Intensive Care Unit forecast description section.
- What makes your tool work effectively? Which types of parameters determine the efficiency of prediction, and which of these parameters are decisive?
Optimal model parameterization is achieved by minimizing the BIC on daily updated data. The forecast error, in terms of squared error score, is plotted and calculated on increasing fractions of the historical series. Figure 7 shows that the error stabilizes downwards after six months of pandemic data. Specifications have been included in the Intensive Care Unit forecast description section.
- Please improve your written. For example, the reference should write in a suitable place, not at the beginning of a sentence.
Done

Reviewer 2 Report
The pandemic outbreak of COVID-19 has posed several questions about public health emergency risk communication. Due to the effort required for the population to adopt appropriate behaviors in response to the emergency, it is essential to inform the public of the epidemic situation with transparent data sources.
The authors participated to the COVID-19ita project (described in the study) aimed to develop a public open-source tool to provide timely, updated information on the pandemic’s evolution in Italy.
It consists of a web-based application.
The web application pulls the data from the official repository of the Italian COVID-19 outbreak at the national, regional, and provincial levels.
The app allows the user to select information to visualize data in an interactive environment and compare epidemic situations over time and across different Italian regions.
At the same time, it provides insights about the outbreak that are explained and commented upon to yield reasoned, focused, timely, and updated information about the outbreak evolution.
The study is very interesting and has the possibility of making an important contribution to the scientific literature of the sector.
The following major improvements are needed:
1) Enter a clear purpose. The last two sentences of the introduction are partly dedicated to this and partly (the second) dedicated to the methods and must be perfected
2) The methods appear as a list of things. Please introduce better what you mean in a structured way
3) Insert a method support flow chart.
4) The results should be refined with a brief introduction of what is going to be said.
5) The figures should be improved as resolution.
6) The conclusions are not true conclusions but future goals.
Author Response
We would like to thank the reviewers for their comments that helped to improve the manuscript quality. The rebuttal letter has been prepared with point-by-point responses. The active track has been turned on in the manuscript to highlight the changes
Reviewer 2
The pandemic outbreak of COVID-19 has posed several questions about public health emergency risk communication. Due to the effort required for the population to adopt appropriate behaviors in response to the emergency, it is essential to inform the public of the epidemic situation with transparent data sources.
The authors participated to the COVID-19ita project (described in the study) aimed to develop a public open-source tool to provide timely, updated information on the pandemic’s evolution in Italy.
The study is very interesting and has the possibility of making an important contribution to the scientific literature of the sector.
The following major improvements are needed:
- Enter a clear purpose. The last two sentences of the introduction are partly dedicated to this and partly (the second) dedicated to the methods and must be perfected.
We agree with the reviewer the sentence has been rewritten.
- The methods appear as a list of things. Please introduce better what you mean in a structured way.
A structured section in the methods was added as suggested by the reviewer.
- Insert a method support flow chart.
We agree with the reviewer, supporting flowcharts have been added in the supplementary material.
- The results should be refined with a brief introduction of what is going to be said.
We agree, the introductory section has been added to the results
- The figures should be improved as resolution.
Agreed, poorly visible images have been improved
- The conclusions are not true conclusions but future goals.
The conclusions have been rearranged and a future research developments section below the discussion has been included.

Round 2
Reviewer 1 Report
In this round, I only check how the authors response my comments. It seems that authors revised manuscript according to my suggestions, and I recommend it being accepted.
Reviewer 2 Report
The manuscript has improved a lot.
The authors have fully replied to my comments.
There are no further requests